# *Wolbachia* and *Asaia* Distribution among Different Mosquito Vectors Is Affected by Tissue Localization and Host Species

**DOI:** 10.3390/microorganisms12030545

**Published:** 2024-03-08

**Authors:** Mahdokht Ilbeigi Khamseh Nejad, Alessia Cappelli, Claudia Damiani, Monica Falcinelli, Paolo Luigi Catapano, Ferdinand Nanfack-Minkeu, Marie Paul Audrey Mayi, Chiara Currà, Irene Ricci, Guido Favia

**Affiliations:** 1School of Biosciences and Veterinary Medicine, University of Camerino, Via Gentile III da Varano, 62032 Camerino, Italy; mahdokht.ilbeigi@unicam.it (M.I.K.N.); monica.falcinelli@unicam.it (M.F.); paololuigi.catapano@unicam.it (P.L.C.); paulaudrey.mayi@unicam.it (M.P.A.M.); 2School of Biosciences and Veterinary Medicine, University of Camerino, CIRM Italian Malaria Network, Via Gentile III da Varano, 62032 Camerino, Italy; alessia.cappelli@unicam.it (A.C.); claudia.damiani@unicam.it (C.D.); irene.ricci@unicam.it (I.R.); 3Department of Biology, The College of Wooster, 1189 Beall Avenue, Wooster, OH 44691, USA; fnanfackminkeu@wooster.edu; 4Institute of Molecular Biology and Biotechnology, Foundation for Research and Technology, Nikolaou Plastira 100, 70013 Heraklion, Greece; chiara.curra@iss.it; 5Department of Infectious Diseases, Istituto Superiore di Sanità, 00161 Rome, Italy

**Keywords:** *Asaia*, *Wolbachia*, mosquito, symbiosis

## Abstract

Microbial communities play an important role in the fitness of mosquito hosts. However, the factors shaping microbial communities in wild populations, with regard to interactions among microbial species, are still largely unknown. Previous research has demonstrated that two of the most studied mosquito symbionts, the bacteria *Wolbachia* and *Asaia*, seem to compete or not compete, depending on the genetic background of the reference mosquito host. The large diversity of *Wolbachia*–*Asaia* strain combinations that infect natural populations of mosquitoes may offer a relevant opportunity to select suitable phenotypes for the suppression of pathogen transmission and for the manipulation of host reproduction. We surveyed *Wolbachia* and *Asaia* in 44 mosquito populations belonging to 11 different species of the genera *Anopheles*, *Aedes*, and *Culex* using qualitative PCR. Through quantitative PCR, the amounts of both bacteria were assessed in different mosquito organs, and through metagenomics, we determined the microbiota compositions in some selected mosquito populations. We show that variation in microbial community structure is likely associated with the species/strain of mosquito, its geographical position, and tissue localization. Together, our results shed light on the interactions among different bacterial species in the microbial communities of mosquito vectors, and this can aid the development and/or improvement of methods for symbiotic control of insect vectors.

## 1. Introduction

Microbial interactions within mosquitoes of different species can have significant effects both on the physiology of the host, and more generally on their biology, and on their susceptibility to pathogens. At the same time, these interactions can condition the effectiveness of control methods based on symbionts, the so-called symbiotic control (SC) [1,2]. From this point of view, little is known about how these interactions can interfere with the efficacy of different SC methods.

Much of this is because the existing studies were mainly conducted using laboratory-bred mosquito populations, which, obviously, can also be very significantly different from field mosquitoes due to the composition of their microbiota [3].

Two bacteria have attracted great interest for their potential in the control of mosquito-borne diseases (MBDs): *Wolbachia*, obligate intracellular bacteria found in many insect species, and *Asaia*, Gram-negative bacteria also widely distributed in insects [4,5,6].

The potential of *Wolbachia* in controlling MBDs has been proven for years, having been corroborated by numerous studies. In fact, the “forced” introduction of some strains of *Wolbachia* in *Aedes aegypti* has shown great efficacy in reducing dengue virus and its competence for transmission [7,8,9,10]. It has also been consistently demonstrated that *Ae. aegypti*, being resistant to dengue virus infection, is capable of rapidly displacing natural/susceptible populations. Consequently, dengue control with *Wolbachia*-based strategies is still ongoing today in various regions of the world [8,9,10,11,12]. Similar approaches have also been proposed for other mosquito species, including some malaria vectors [13,14].

Concerning *Asaia*, its strict ecological association with many different mosquito species has attracted much interest in the frame of the paratransgenic control of malaria and other MBDs. Indeed, *Asaia* may infect most of the members of a population if not all, including all developmental stages and several anatomical districts, thus acclaiming itself as one of the best paratransgenic agents. In this frame, at the laboratory level, paratransgenic strains of *Asaia* that inhibit malaria transmission have been produced [15,16]. Moreover, *Asaia* can stimulate the basal level of mosquito immunity to naturally reduce the development of malaria parasite oocysts in *Anopheles stephensi*, thus expanding its potential in SC approaches, not only through paratransgenesis, but also as a potential effector for insect immune priming [17,18]. The establishment of *Wolbachia* in the host mosquito can be inhibited by the co-presence of other bacteria, which could limit its effectiveness in controlling some MBDs [19]. In some genetic backgrounds, the presence of *Wolbachia*, and therefore its potential control efficacy, is related to the absence/presence of *Asaia* strains. A possible explanation for this competition could lie in the fact that *Asaia* and *Wolbachia* potentially compete for the same resources, but this still needs to be better clarified. Nevertheless, experimental evidence of a competition between the two symbionts has been reported in mosquitoes of the genus *Anopheles* and in *Ae. aegypti* [20,21,22]. In order to be able to evaluate the effectiveness of control methods based on symbionts, and, in this specific case, on *Asaia* and *Wolbachia*, a detailed study of the distribution of these two bacteria in different species/strains of mosquitoes coming from regions characterized by different eco-ethological contexts has been performed. Indeed, the large diversity of *Wolbachia*–*Asaia* strain combinations able to infect natural populations of mosquitoes may offer a relevant opportunity to select suitable phenotypes for the suppression of pathogen transmission and for the manipulation of host reproduction. Here, we surveyed *Wolbachia* and *Asaia* in 44 mosquito populations using molecular tools, showing that variation in microbial community structure is likely associated with several factors like mosquito species/strains, localization within the mosquito tissues, and the geographical localization of the mosquito.

## 2. Materials and Methods

### 2.1. Mosquito Collection and Identification

Mosquitoes were collected in four locations in 2022 and 2023: Italy, Cameroon (Africa), Crete (Greece), and Ohio (USA) (Appendix A).

#### 2.1.1. European Collections

In Italy, *Aedes koreicus* and *Aedes. japonicus* were collected in Feltre, Pedavena, Sospirolo, and Alano di Piave (Veneto region), whereas *Aedes albopictus* were collected in Sospirolo, Pedavena, Feltre, and Petriolo (Marche region). Mosquito larvae were mainly collected from artificial containers by dipping and delivered to the laboratory in a box fridge. Mosquitoes were identified morphologically according to Montarsi et al., 2013 [23] and molecularly as described in Schneider et al., 2016 [24]. *Culex pipiens* mosquitoes were collected in Camerino (Marche), Pedavena (Veneto), and Pisa (Toscana). The mosquito genera were evaluated using morphological keys [25], whereas the species were molecularly identified as described in Fotakis et al., 2022 [26].

In Crete, *Ae. albopictus* specimens were collected in Gazi, Heraklion, Rethymnon, and Hersonissos; *Cx. pipiens* specimens were collected in Rethymnon, Heraklion, and Hersonissos, using BG-Sentinel 2 traps (Blue line S.r.l, Forlì, Italy). The mosquito genera were evaluated using morphological keys [25], whereas the species were molecularly identified as described in Fotakis et al., 2022 [26].

#### 2.1.2. African Collections

In Cameroon, *Anopheles gambiae*, *Anopheles coluzzii*, *Anopheles arabiensis*, *Anopheles pharoensis*, *Anopheles ziemanni*, and *Ae. albopictus* were collected from 7 locations belonging to 5 different eco-geographical areas (Santchou, Yaoundé, Adamaoua, Douala, Mbandjock, Mbalmayo, and Yangah). Mosquitoes were collected using Center for Disease Control Light Traps, Human Landing Catches, and Prokopack aspirators. Adults were identified morphologically using the identification keys [27,28]. *An. gambiae* complex specimens were identified using the SYBR green-based assay described by Chabi et al., 2019 [29]. *An. phaorensis* and *An. ziemanni* samples were sequenced at cytochrome C oxidase subunit 1 (COI) loci for species confirmation [30].

A cohort of *Ae. aegypti* and *Anopheles funestus* collected in Ouagadougou (Burkina Faso) in 2008 were included in the study.

#### 2.1.3. USA Collections

In Ohio, *Ae. japonicus* mosquitoes were collected in Wooster city using gravid traps baited with yeasts. Mosquitoes were morphologically and molecularly identified as described in Nanfack-Minkeu et al., 2023 [31].

### 2.2. DNA Extraction

Before the DNA extraction, the insect surface was sterilized in 70% ethanol and rinsed for three times in sterile PBS. Samples were homogenized with sterile 1 mm wide glass beads (Next Advance Inc., New York, NY, USA) for 30 s at 6800 rpm with an automatic tissue homogenizer (Precellys 24, Bertin Instrument, Montigny-le-Bretonneux, France). Genomic DNA was extracted using a JetFlex Genomic DNA Purification kit (Invitrogen, Thermo Fisher Scientific, Waltham, MA, USA) according to the manufacturer’s instructions. The concentration and purity of the DNA was determined using a NanoDrop™ spectrophotometer (Thermo Scientific, Waltham, MA, USA). Finally, the DNA was stored at −20 °C prior to analysis.

### 2.3. Wolbachia and Asaia Detection

A total of 1098 mosquitoes were tested to detect the presence of the bacteria *Asaia* and *Wolbachia* using specific oligonucleotides (Table 1). For *Wolbachia*, a semi-nested PCR targeting the *Wolbachia* 16S rRNA gene was performed using 50 ng genomic DNA, 1X Buffer, 0.25 mM dNTPs, 0.9U DreamTaq Polymerase (Thermo Scientific, Waltham, MA, USA), 240 nM Wol-For, 160 nM Wol-rev2, and 120 nM Wol-rev3 [21]. The amplification cycle consisted of an initial denaturation at 95 °C for 3 min, followed by 5 cycles consisting of denaturation at 95 °C for 30 s, annealing at 54 °C for 30 s, and extension at 72 °C for 30 s, and 25 cycles consisting of denaturation at 95 °C for 30 s, annealing at 52 °C for 30 s, and extension at 72 °C for 30 s, concluding with a final extension step of 10 min at 72 °C. For *Asaia*, specific PCR targeting of the 16S rRNA gene was performed using 50 ng genomic DNA, 1X Buffer, 0.25 mM dNTPs, 0.9U DreamTaq Polymerase (Thermo Scientific, Waltham, MA, USA), and 200 nM of *Asaia*NewFor and *Asaia*NewRev oligonucleotides [5]. The amplification protocol included initial denaturation at 95 °C for 3 min, followed by 30 cycles consisting of denaturation at 95 °C for 30 s, annealing at 60 °C for 30 s, and extension at 72 °C for 30 s, concluding with a final extension step of 10 min at 72 °C. The PCR products were electrophoresed on a 1% agarose gel to verify the presence and size of the specific amplicons.

### 2.4. Metagenomics Analysis

For each of the species analyzed, the microbiota profiles of 10 single mosquitoes were measured. *Ae. japonicus*, collected in Italy and Ohio, and *Ae. albopictus* and *Cx. pipiens*, collected in Italy and Crete, were analyzed through NGS analysis using the bacterial target 16S rRNA gene. Blank extractions were analyzed as negative controls to evaluate possible bacterial contaminations occurring during the template preparation. Moreover, 16S rRNA gene profiling was conducted with SYNBIOTEC srl (Camerino, Italy). Library preparation was performed by covering the hypervariable region V3-V4 of the 16S rRNA gene using the oligonucleotides 341F and 785R [33]. The data were pre-processed using the Illumina MiSeq—2 × 250 PE—V2 nano, and the readings were sorted by amplicon in line barcodes. No quantifiable libraries were produced from the negative controls.

Quality control, taxonomic attribution and diversity, and abundance analyses were performed in Qiime2 version 2023.5 [34]. Qiime tools were imported and the “CasavaOneEightSingleLanePerSampleDirFmt” format was used to import the 16S sequences. Readings were merged with the qiime vsearch merge-pair. Quality control and denoising were performed using qiime quality filter q-score and qiime deblur denoise 16S. For the taxonomy identification, Silva138 [35] and qiime feature-classifier classify-sklearn were used. Plots were obtained with qiime taxa barplot and GraphPad (https://www.graphpad.com/) (accessed on 14 February 2024). Rarefaction curves, Shannon alpha diversity, and Bray–Curtis PCoA were obtained with qiime diversity alpha-rarefaction, qiime diversity alpha-group-significance, and qiime emperor plot. The data presented in this study are deposited in the NCBI repository (accession number PRJNA1069599).

### 2.5. Asaia and Wolbachia Quantification via qPCR

*Asaia* and *Wolbachia* density was evaluated in organs (male and female guts, male and female reproductive organs, and female salivary glands) of *Ae. albopictus* (collected in Petriolo, Marche region), *Cx. pipiens* (collected in Camerino, Marche region), and *An. stephensi* (laboratory colony) via qPCR. Eight pools of 10 organs were obtained by dissecting 15-day-old mosquitoes in a drop of sterile 1× PBS using sterile needles under a stereomicroscope (Olympus, Tokyo, Japan). Samples were homogenized and the DNA was extracted as described above. PCR assays were performed using 1XHOT FIREPol^®^ EvaGreen^®^ qPCR Supermix (Tartu, Estonia), 200 nM of oligonucleotides, and 50 ng of genomic DNA. Specific oligonucleotides targeting the 16S DNA were used to quantify *Asaia* and *Wolbachia.* Moreover, the genes Ae-rps7, Cx-rps3, and As-rps7 were amplified as housekeeping genes (for *Ae. albopictus*, *Cx. pipiens*, and *An. stephensi*, respectively) [21,32]. All gene sequences are summarized in Table 1. Reactions were run on a CFX thermocycler (Bio-Rad, Hercules, CA, USA) using the following cycling conditions: 1 cycle of 95 °C for 12 min, 40 cycles of 95 °C for 1 min, 60 °C for 1 min, and 74 °C for 30 s. The melting peak for each target was obtained with the following steps: 65 °C to 95 °C for 5 s with an increment of 0.5 °C. The quantity of amplified targets was measured using standard curves obtained by eight serial dilutions of specific plasmids for each amplicon (from 2 to 2 × 10^−7^ ng). The standard curves used in the experiments had the following parameters (E = efficiency; R^2^ = correlation coefficient): *Asaia:* E = 91.8%, R^2^ = 0.9983; *Wolbachia:* E = 91.2%, R^2^ = 0.997; Ae-rps7: E = 99.5, R^2^ = 0.9996; Cx-rps3: E = 100.6, R^2^ = 0.9985; and As-rps7: E = 96.2, R^2^ = 0.999. Amounts of *Asaia* and *Wolbachia* were estimated as relative quantities, calculating the number gene copy ratio (number gene copy of 16S rRNA/number gene copy of housekeeping gene). The amounts of *Asaia* and *Wolbachia* were estimated using the Bio-Rad CFX Maestro Software 2.3 (version 5.3.022.1030) and the GraphPad software (http://www.graphpad.com) (accessed on 16 February 2024). Data were obtained from the average of eight pools per organ. The value of each pool resulted from the average of two technical replicates that were compared through the Mann–Whitney test.

## 3. Results

### 3.1. Asaia–Wolbachia Distribution in Different Mosquito Populations

Specific PCR tests were used to verify the circulation of the two symbionts in different species and populations of mosquitoes belonging to the genera *Anopheles*, *Aedes*, and *Culex*.

Concerning the genus *Anopheles*, *Asaia* was detected in all species and populations tested, while *Wolbachia* was never detected (Table 2). The relative prevalence of *Asaia* is highly variable and seems to be dependent on the geographical region of Cameroon in which the mosquitoes were collected. Indeed, those collected in Yangah and Amadoua show relative prevalence rates spanning from 2.1 to 29.2%, while those collected in other areas of Cameroon show prevalence rates spanning from about 33 to 100%. In *An. funestus* collected in Burkina Faso, *Asaia* was detected in 46.4% of the samples.

Concerning the genus *Aedes*, *Asaia* was detected in all of the populations tested (Table 2). In all *Ae. albopictus* populations, the prevalence of *Wolbachia* was 100%, and the prevalence of *Asaia* ranged from 91 to 100%. In *Ae. aegypti*, *Ae. koreicus*, and *Ae. japonicus*, the prevalences of *Asaia* were 42.9%, 97–100%, and 97.4–100%, respectively, while *Wolbachia* was never detected.

An exception to these data is represented by the circulation of the two symbionts in the populations collected in Greece, on the island of Crete. The populations of *Ae. albopictus* analyzed always revealed the presence of *Wolbachia* in all of the tested insects. As regards *Asaia*, the percentages fluctuated from 0 to 61.5%.

Moreover, in *Ae. japonicus*, the circulation of *Asaia* detected in the Italian populations (from 97.4 to 100%) was much higher than that recorded in the American population (49%). In consideration of this, two comparative metabarcoding analyses were conducted: the first compared the Italian *Ae. albopictus* with its Greek counterpart; the second compared *Ae. japonicus* collected in the field in Italy versus in the United States. Both analyses revealed that the composition of the microbiota varied between the compared populations with a more than likely impact on the circulation of *Asaia*. Nevertheless, while the microbiota of the Italian and Greek populations of *Ae. albopictus* were dominated by the presence of *Wolbachia*, and therefore their differences concern a quantitative minority share of the microbiota represented by various bacterial species (Figure 1A), the microbiota of the Italian and North American populations of *Ae. japonicus* differed markedly. In fact, while *Asaia* was largely the predominant bacterium in the Italian population, *Pseudomonas* prevailed in the American population (Figure 1B).

Concerning the genus *Culex*, both *Asaia* and *Wolbachia* were detected in all of the Italian and Greek populations of *Cx. pipiens* tested (Table 2). In these populations, the prevalence of *Wolbachia* was 100% in all of the populations, while *Asaia* was found in 100% of the Italian mosquitoes, with a range of 36.4 to 54.5% in the Greek ones.

Similarly to the findings in some *Aedes* populations, the circulation of *Asaia* detected in the Italian populations of *Cx. pipiens* was higher than that recorded in the Greek populations. Thus, comparative metagenomic analyses were conducted, revealing that the microbiota associated with the two populations differed very substantially: while in the Italian population *Wolbachia* was definitely the predominant bacterium in all of the samples analyzed, confining *Asaia* to a limited quantity, in the Greek population, the situation was much more varied, with *Wolbachia* being confined to decidedly more modest proportions and a greater richness of other bacteria such as *Pseudomonas* and *Acinetobacter* (Figure 1C).

Regarding the 16S results, the principal coordinates analysis (PCoA) confirmed both the large differences between mosquitoes collected in Italy compared to those collected in Ohio and Crete, and the strong similarity of the microbial composition in *Ae. albopictus* and *Cx. pipiens* collected in Italy. Furthermore, the high quantity of *Asaia* in the Italian population of *Ae. japonicus* appears to have affected the entire microbial community enough to isolate this population from the others (Appendix A).

### 3.2. Competition in Different Mosquito Organs

The coexistence of *Wolbachia* and *Asaia* was found only in *Cx. pipiens* and *Ae. albopictus*; therefore, we verified in which anatomical organs the competition between these two symbionts occurred through quantitative PCR. As shown in Figure 2, the comparative analysis of male and female guts, reproductive organs, and female salivary glands clearly indicated that competition occurs only in the reproductive organs of both sexes. In these organs, in both sexes and in both species, the presence of *Wolbachia* was much greater than that of *Asaia*.

Quantitative PCR was also used to compare the amount of *Asaia* in the reproductive organs and guts of mosquitoes with *Wolbachia* (*Ae. albopictus*) and without *Wolbachia* (*An. stephensi*). We recorded a notably greater presence of *Asaia* in both the guts and reproductive organs of both sexes in *An. stephensi* (Figure 3).

## 4. Discussion

We have previously demonstrated a role for *Wolbachia* in preventing some mosquito species from conducting a stable and successful *Asaia* infection in the gonads [21]. It has also been proposed that *Asaia* plays a role in the difficulty of *Wolbachia* to infect anopheline mosquitoes [20,21].

Therefore, in this study, we analyzed mosquitoes from different geographical contexts to analyze a large diversity of mosquito populations in order to better understand the circulations of the two symbiotic bacteria in different host vectors.

In all of the anopheline populations analyzed, we never found the presence of *Wolbachia*. This is not surprising, since although *Wolbachia* has been reported to be present in some wild populations of anophelines, the infection frequencies are very low [36,37,38,39]. On the other hand, we found *Asaia* in all of these populations, albeit with variable infection frequencies that appeared to be strongly correlated to the geographical location of the mosquito population analyzed. For instance, we analyzed several anopheline populations collected in seven different areas in Cameroon. Interestingly, in the populations collected in two areas located in the north of the country, the circulation of *Asaia* was much lower than those recorded in populations collected in other areas of the nation. This would seem to suggest a circulation of *Asaia* somehow conditioned by eco-ethological factors and not only by the host species.

In the analyzed *Aedes* species, the situation was substantially different and much more varied. In *Ae. aegypti*, *Ae. koreicus*, and *Ae. japonicus*, we did not record the presence of *Wolbachia*, while *Asaia* was always present, albeit with significant fluctuations. These data are consistent with the literature; in fact, for *Ae. aegypti*, *Ae. koreicus*, and *Ae. japonicus*, there have only been episodic and very rare reports of *Wolbachia* infections [40,41,42,43].

On the other hand, the circulation of *Asaia* in these species has been strongly described [41,44,45,46]. Nonetheless, the comparison of the circulation of *Asaia* in North American and Italian populations of *Ae. japonicus* revealed very different frequencies, being significantly higher in the Italian ones.

The comparative metagenomic analysis between an Italian and a North American population highlighted that in the Italian population, *Asaia* was the dominant bacterium, while in that of Ohio, the large presence of *Pseudomonas* and *Pantoea* relegated *Asaia* in very small amounts, confirming the different flows of microbial competition in these different host populations.

For *Ae. albopictus*, very high infection rates of both bacteria have been confirmed and reported in many tested populations [47,48,49], and a similar situation was detected in the *Culex* populations we analyzed in this study.

Nonetheless, even in the Greek populations of *Ae. albopictus* and *Cx. pipiens*, the circulation of *Asaia* was found to be lower than in the Italian populations. The comparative metagenomic analysis between the Italian and Greek populations once again highlighted that in the Italian population, although *Wolbachia* was the dominant bacterium, *Asaia* was still present in most of the individuals, while in the Greek populations, the different microbiota composition translated into very small quantities of *Asaia* circulating in the host, thus confirming different patterns of microbial competition in these different host populations.

Nevertheless, these species offered us the chance to better define the kind of competition that occurs between these two symbionts: dissecting the scale of this competition at the tissue level shows that competition occurs mainly in the reproductive organs, in which the high quantity of *Wolbachia* seems to strongly limit the circulation of *Asaia*.

Indeed, the comparison of *Asaia* circulation between *An. stephensi* and *Ae. albopictus* seems to present a further demonstration of the mutualistically exclusive relationship between the two symbionts. In *An. stephensi*, in which *Wolbachia* was absent, the amounts of *Asaia* found were much higher than those found in *Ae. albopictus*, both in the guts and in the reproductive organs.

## 5. Conclusions

The circulation of *Wolbachia* and *Asaia* in mosquito vectors appears to be conditioned by different factors such as the host species, the geographical location of the vector, and the reference tissue of the mosquito. Although we cannot exclude other factors, such as the genetic characteristics of different strains of the symbionts and the eco-ethological contexts of reference of the mosquito, our study is further evidence of the competitive relationship between *Asaia* and *Wolbachia* in many mosquito vectors. Since they represent two of the most used and/or proposed symbionts for symbiotic control methods, our evidence represents a relevant contribution to understanding the *Wolbachia*–*Asaia* strain combinations able to infect natural populations of mosquitoes, aiming to select suitable phenotypes for the suppression of pathogen transmission and for the manipulation of host reproduction.

## Figures and Tables

**Figure 1 microorganisms-12-00545-f001:**
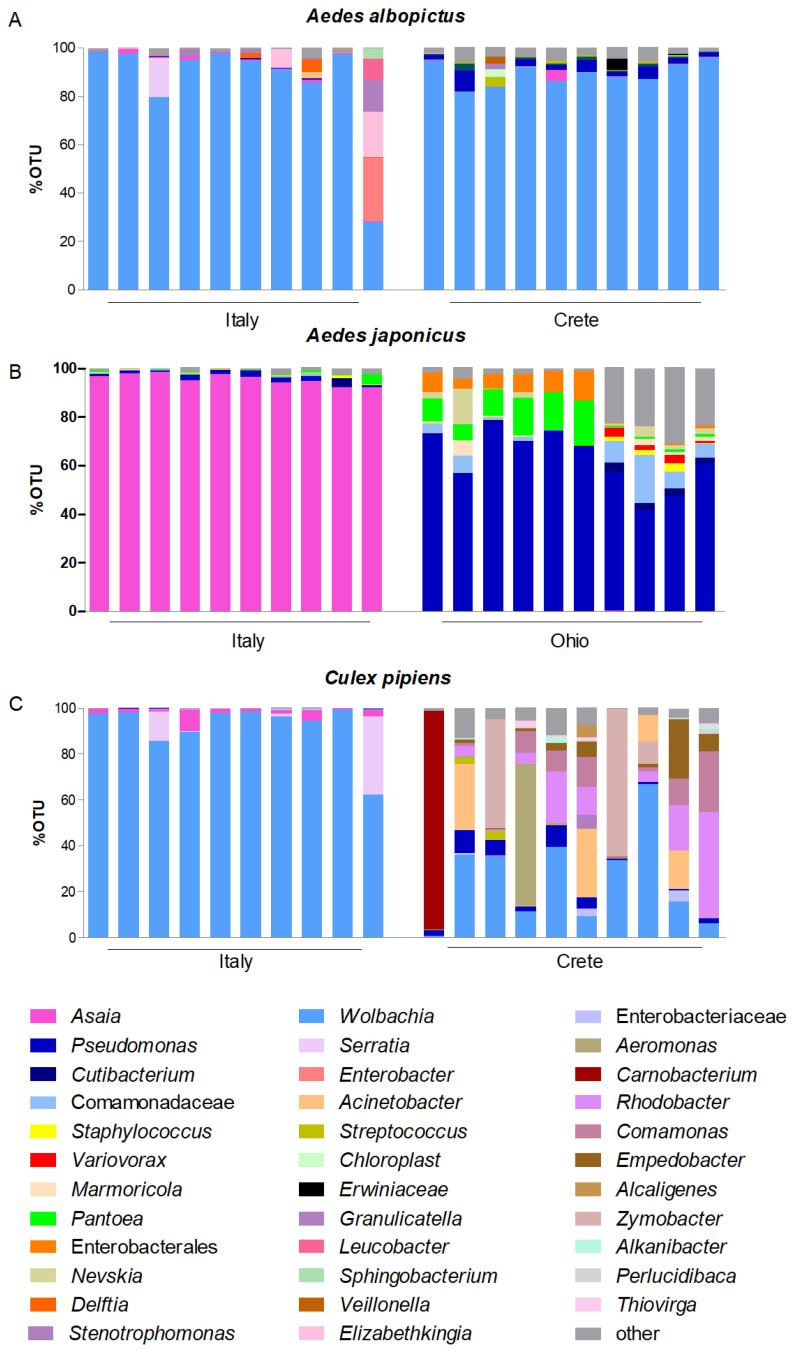
Stacked bar plots showing the relative abundances of bacterial taxa distributed among mosquito species: (**A**) *Aedes albopictus*; (**B**) *Ae. japonicus*; (**C**) *Cx. pipiens*. X-axis indicates the mosquito samples and Y-axis indicates the relative abundance of bacterial taxa calculated as a percentage of the Operational Taxonomic Units (OTUs). Only OTUs > 2% of the total readings are represented.

**Figure 2 microorganisms-12-00545-f002:**
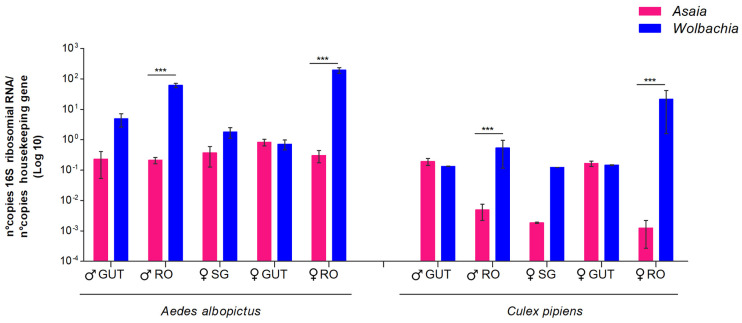
Quantitative detection of *Asaia* and *Wolbachia* in organs of *Ae. albopictus* and *Cx. pipiens* mosquito species obtained via qPCR. The relative amount of the bacteria is expressed as a ratio of bacterial 16S rRNA and mosquito rps7 (*Ae. albopictus*) or rps3 genes (*Cx. pipiens*) copies in a logarithmic scale. Abundance results from the mean ± SEM of eight pools (10 organs) for each species. Statistically significant differences are represented by asterisks (*p* < 0.001), as determined through multiple comparisons using the Mann-Whitney test. ♀: female; ♂: male; RO: reproductive organs; SG: salivary glands.

**Figure 3 microorganisms-12-00545-f003:**
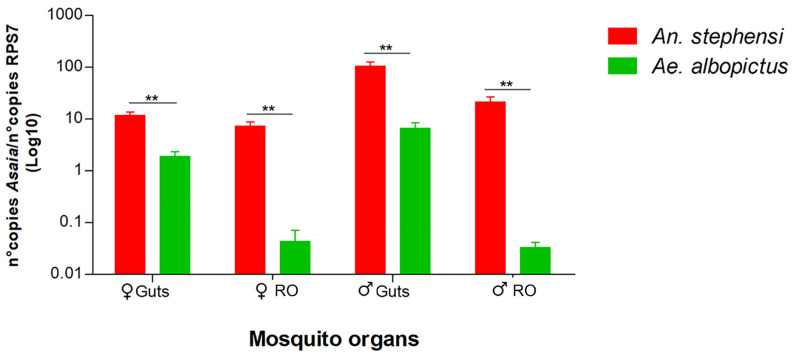
Quantitative detection of *Asaia* in organs of *An. stephensi* and *Ae. albopictus* mosquito species obtained via qPCR. The relative amount of the bacteria is expressed as a ratio of bacterial 16S rRNA and mosquito rps7 genes (*An. stephensi* and *Ae. albopictus*) copies in a logarithmic scale. Abundance results from the mean ± SEM of eight pools (10 organs) for each species. Statistically significant differences are represented by asterisks (*p* < 0.01), as determined through multiple comparisons using the Mann–Whitney test. ♀: female; ♂: male; RO: reproductive organs.

**Table 1 microorganisms-12-00545-t001:** Oligonucleotides used in the study.

Name	Target	Forward Oligonucleotides (5′→3′)	Reverse Oligonucleotides (5′→3′)	Reference
*Wolbachia*	16S rRNA	GAAGATAATGACGGTACTCAC	R2: GTCAGATTTGAACCAGATAGA R3: GTCACTGATCCCACTTTAAATAAC	[21]
*Asaia*New	16S rRNA	GCGCGTAGGCGGTTTACAC	AGCGTCAGTAATGAGCCAGGTT	[5]
*Asaia* qPCR	16S rRNA	TAGCGTTGCTCGGAATGACTGG	CGTATCAAATGCAGCCCCAAGG	[32]
*Wolbachia* qPCR	16S rRNA	GAAGATAATGACGGTACTCAC	CCTACGCGCTCTTTACGCCCA	This work
*Ae-rps7*	RPS7	CGCGCTCGTGAGATCGA	GCACCGGGACGTAGATCA	[21]
*Cx-rps3*	RPS3	AGCGTGCCAAGTCGATGAG	ACGTACTCGTTGCACGGATCTC	[21]
*As-rps7*	RPS7	AGCAGCAGCAGCACTTGATTTG	TAAACGGCTTTCTGCGTCACCC	[32]

**Table 2 microorganisms-12-00545-t002:** Samples analyzed for *Asaia* and *Wolbachia*.

Mosquito Species	Collection Site	Country	*Asaia* % (Positive/Total)	*Wolbachia* % (Positive/Total)
*An. arabiensis*	Yangah	Cameroon	9/37 (24.3)	0/37 (0)
*An. coluzzii*	Yangah	Cameroon	1/10 (10)	0/10 (0)
	Yaoundé	Cameroon	33/39 (84.6)	0/39 (0)
	Douala	Cameroon	20/38 (52.6)	0/38 (0)
	Mbalmayo	Cameroon	34/34 (100)	0/34 (0)
*An. funestus*	Ouagadougou	Burkina Faso	32/69 (46.4)	0/69 (0)
*An. gambiae*	Santchou	Cameroon	15/45 (33.3)	0/45(0)
	Yaoundé	Cameroon	16/25 (64)	0/25 (0)
	Adamaoua	Cameroon	1/48 (2.1)	0/48 (0)
	Douala	Cameroon	13/25 (52)	0/25 (0)
	Mbandjock	Cameroon	41/41 (100)	0/41(0)
	Mbalmayo	Cameroon	5/5 (100)	0/5 (0)
*An. pharoensis*	Yangah	Cameroon	14/48 (29.2)	0/48 (0)
*An. ziemanni*	Yangah	Cameroon	9/48 (18.8)	0/48 (0)
*Ae. aegypti*	Ouagadougou	Burkina Faso	3/7 (42.9)	0/7 (0)
*Ae. albopictus*	Petriolo	Italy	72/72 (100)	72/72 (100)
	Sospirolo	Italy	11/12 (91.6)	12/12 (100)
	Pedavena	Italy	11/11 (100)	11/11 (100)
	Feltre	Italy	1/1 (100)	1/1 (100)
	Dschang	Cameroon	44/44 (100)	44/44 (100)
	Gazi	Crete	0/5 (0)	5/5 (100)
	Heraklion	Crete	1/8 (12.5)	8/8 (100)
	Rethymnon	Crete	9/21 (42.9)	21/21 (100)
	Hersonissos	Crete	4/26 (15.4)	26/26 (100)
*Ae. koreicus*	Sospirolo	Italy	32/33 (97)	0/33 (0)
	Pedavena	Italy	52/52 (100)	0/52 (0)
	Feltre	Italy	20/20 (100)	0/20 (0)
	Alano di Piave	Italy	24/24 (100)	0/24 (0)
*Ae. japonicus*	Sospirolo	Italy	20/20 (100)	0/20 (0)
	Pedavena	Italy	29/29 (100)	0/29 (0)
	Feltre	Italy	37/38 (97.4)	0/38 (0)
	Wooster	Ohio (USA)	19/39 (48.7)	0/39 (0)
*Cx. pipiens*	Camerino	Italy	38/38 (100)	38/38 (100)
	Pedavena	Italy	2/2 (100)	2/2 (100)
	Pisa	Italy	42/42 (100)	42/42 (100)
	Rethymnon	Crete	6/11 (54.5)	11/11 (100)
	Heraklion	Crete	4/9 (44.4)	9/9 (100)
	Hersonissos	Crete	8/22 (36.4)	22/22 (100)

## Data Availability

All of the readings related to the 16S Miseq analysis (Bioproject PRJNA1069599) have been deposited in the EMBL Nucleotide Sequence Database (NCBI).

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
