# Peer review of "Wolbachia and Asaia Distribution among Different Mosquito Vectors Is Affected by Tissue Localization and Host Species"

_microorganisms, 2024, doi:10.3390/microorganisms12030545_

Round 1

Reviewer 1 Report

Comments and Suggestions for Authors

The manuscript by Khamseh Nejad deals with an interesting topic related to pest control: symbiont surveys in mosquitoes of importance in order to develop future symbiotic control. Several species and populations were studied and when it was justified, deeper analyses were accomplished. The work is well done and the manuscript, very well written, is almost flawless. A few concerns must be clarified before publication, which I recommend.

Line 91: All species must be written in full when introduced for the first time (e.g. Aedes was abbreviated in Ae. japonicas and Ae. albopictus because Aedes was already alluded in Ae. aegipty, I think). Also, if the journal requires so, the author name hsoudl be written when the species appears for the first time in the manuscript.

Line 96: Following a stop or a period, the genus must be written in full even when it was already alluded before in the text.

Lines 98 and 103: replace “molecular” by “molecularly”.

Line 111: The same as line 96 for Anopheles.

Line 149: replace “16S” by “16S rRNA”.

Regarding the experimental design of the 16s rRNA profiling, how many samples were assayed? Any pooling?

Lines 172-173: clarify the abbreviations used below in Figures 2 and 3 (for example: reproductive organs (RO), salivary glands (SG)).

Italicize all the scientific names along the whole text.

Table 1 legend: replace “e” by “and”.

In table 1, I recommend %(positive/total).

Have the authors noticed any correlation between geographic location and prevalence? I think a statistical test could be done.

Lines 216-217: In fact, the exception is the coinfection.

Line 222: replace “metagenomic” by “metabarcoding”. The authors collected 16S rRNA sequences, not genomes of the bacterial community.

Lines 300-302: Is any information available about possible genetic differentiation between those host populations?

Lines 320-335: I think those three paragraphs are in fact only one. The stops interrupt the connecting thread for the reader.

Author Response

We thank both reviewers for their constructive criticisms and suggestions that have helped us to improve the overall quality of the manuscript.

Reviewer 1

The manuscript by Khamseh Nejad deals with an interesting topic related to pest control: symbiont surveys in mosquitoes of importance in order to develop future symbiotic control. Several species and populations were studied and when it was justified, deeper analyses were accomplished. The work is well done and the manuscript, very well written, is almost flawless. A few concerns must be clarified before publication, which I recommend.

Line 91: All species must be written in full when introduced for the first time (e.g. Aedes was abbreviated in Ae. japonicas and Ae. albopictus because Aedes was already alluded in Ae. aegipty, I think). Also, if the journal requires so, the author name hsoudl be written when the species appears for the first time in the manuscript.

We modified the text accordingly to the reviewer's suggestion.

Line 96: Following a stop or a period, the genus must be written in full even when it was already alluded before in the text.

We modified the text accordingly to reviewer's suggestion.

Lines 98 and 103: replace “molecular” by “molecularly”.

We modified the text accordingly to reviewer's suggestion.

Line 111: The same as line 96 for Anopheles.

We modified the text accordingly to reviewer's suggestion.

Line 149: replace “16S” by “16S rRNA”.

We modified the text accordingly to reviewer's suggestion.

Regarding the experimental design of the 16s rRNA profiling, how many samples were assayed? Any pooling?

Following reviewer’s suggestion, we have modified the section 2.4 of M&M indicating the number of single mosquitoes analyzed in 16S NGS (Line: 161).

Lines 172-173: clarify the abbreviations used below in Figures 2 and 3 (for example: reproductive organs (RO), salivary glands (SG)).

We modified both Fig. 2 and 3 legends accordingly to reviewer suggestions.

Italicize all the scientific names along the whole text.

We italicized all scientific names through the manuscript.

Table 1 legend: replace “e” with “and”.

We modified the text accordingly to reviewer's suggestion.

In table 1, I recommend %(positive/total).

We modified the text accordingly to reviewer's suggestion.

Have the authors noticed any correlation between geographic location and prevalence? I think a statistical test could be done.

Asaia is a very widespread bacteria in insect populations, although a difference in prevalence in some of populations collected was recorded, but due to both the limited number of individuals collected in some geographical area and the presence of some species only in specific geographical areas, we were not pushed to apply statistical analysis to this aim. Nevertheless, we have conducted a statistical analysis, trough PCoA graph of the 16S rRNA Miseq, only for those mosquito species recorded in different geographical areas, thus showing a correlation between microbiota biodiversity and geographical location (as in the case of Aedes japonicus collected in Italy vs those collected in Ohio).

Lines 216-217: In fact, the exception is the coinfection.

Yes, we agree with that consideration.

Line 222: replace “metagenomic” by “metabarcoding”. The authors collected 16S rRNA sequences, not genomes of the bacterial community.

We modified the text accordingly to reviewer's suggestion.

Lines 300-302: Is any information available about possible genetic differentiation between those host populations?

To date, although very likely, we have no evidence about a possible implication relative to genetic differentiation between the different species of mosquito hosts, but this is a crucial point to be studied in future.

Lines 320-335: I think those three paragraphs are in fact only one. The stops interrupt the connecting thread for the reader.

We grouped the last 3 paragraphs in just one as suggested by the reviewer.

Reviewer 2 Report

Comments and Suggestions for Authors

The authors of the manuscript aimed to  analyze mosquitoes from different geographical areas to analyze a large diversity of mosquito populations to better understand the circu lations of the Wolbachia and Asaia in different host-vectors.

The topic covered is interesting and the manuscript is well structured. Table S2 could be reported in the text as table 1, in the methods section, also reporting the oligonuleotides used for Wolbachia and Asaia detection mentioned in paragraph 2.3 of the methods.

Author Response

We thank both reviewers for their constructive criticisms and suggestions that have helped us to improve the overall quality of the manuscript.

The authors of the manuscript aimed to  analyze mosquitoes from different geographical areas to analyze a large diversity of mosquito populations to better understand the circu lations of the Wolbachia and Asaia in different host-vectors.

The topic covered is interesting and the manuscript is well structured. Table S2 could be reported in the text as table 1, in the methods section, also reporting the oligonuleotides used for Wolbachia and Asaia detection mentioned in paragraph 2.3 of the methods.

We moved Table S2 in the main text (now Table 1) indicating also the oligonucleotides used for Wolbachia and Asaia detection (mentioned in paragraph 2.3). Consequently, we renumbered the original Table 1 in Table 2 in the present version.